# Cyclic Graph Dynamic Multilayer Perceptron for Periodic Signals

## Abstract

We propose a feature extraction for periodic signals. Virtually every mechanized transportation vehicle, power generation, industrial machine, and robotic system contains rotating shafts. It is possible to collect data about periodicity by measuring a shaft's rotation. However, it is difficult to perfectly control the collection timing of the measurements. Imprecise timing creates phase shifts in the resulting data. Although a phase shift does not materially affect the measurement of any given data point collected, it does alter the order in which all of the points are collected. It is difficult for classical methods, like multi-layer perceptron, to identify or quantify these alterations because they depend on the order of the input vectors' components. This paper proposes a robust method for extracting features from phase shift data by adding a graph structure to each data point and constructing a suitable machine learning architecture for graph data with cyclic permutation. Simulation and experimental results illustrate its effectiveness.

## 1 Introduction

Understanding what phenomena, a rotating shaft is experiencing is critical for machine health monitoring. From industrial manufacturing equipment, transportation systems, to consumer products, rotating shafts are in many mechanical devices. Many issues such as long-term fatigue, wear related issues, and acute failures can cause symptoms that are detectable from the shaft. The effort and flow variables associated with the shaft are desirable state variables to measure in nearly all these cases mentioned. Although these physical variables may provide useful information for detecting anomalies and estimating symptoms, one should extract features hidden in these signals. Therefore, an efficient feature extraction method plays an important role in anomaly detection and symptoms recognition.

Deep learning networks achieved remarkable results compared to the traditional methods. The time-frequency analysis, such as the short-time Fourier transform Xie et al. (2012) and the wavelet transform YanPing et al. (2006); Al-Badour et al. (2011), are well known feature extraction methods. For example, one can detect a certain bending mode by paying attention to the resonance frequency. Namely, domain knowledge expertise is needed to extract features from a time-frequency representation associated with particular phenomena. Some deep convolutional neural network (CNN) architectures achieved good results by taking time-frequency images as inputs Verstraete et al. (2017); Guo et al. (2018). Although many machine learning methods with preprocessing schemes were used to extract signal features, many of them do not really consider the specific characteristics of signals. For example, the output from a general CNN is compressed by pooling regardless of time or frequency direction.

A method that considers the relative order information of signals is necessary. Classical methods, such as multi-layer perceptron (MLP), regard signals as vectors and accordingly use vectors as inputs. However, a vector does not give relationship information that might exist between coordinates. Therefore, while classical methods can measure data points, it is difficult to detect whether they are in proper order relative to each other. This could occur due to pooling even if the signal is converted into an image by some method and input to the CNN. The relative order is very important to classify them. For example, the data obtained from a rotating machinery is periodic as in Figure 5. Figure 5a shows noise associated with the rotation period, and so it is related to a rotation anomaly such as a crack in a gear. However, Figure 5b shows noise that is different from the rotation period, and

thus there is a possibility that it is not related to rotation, but perhaps an abnormality of the sensor system. Even with a classical method, it is possible to classify them if such data are included in the training data. However industrial machines would require significant time and cost to run the necessary experiments for collecting data. If the abnormality to be detected was rare, then the required effort would be magnified.

The proposed method to solve this problem, considers a graph structure for each data point. This scheme provides a relative order information about the vector coordinates. It then applies a graph neural network, such as Atwood & Towsley (2016); Kipf & Welling (2017), to the graph structured data. For example, the relative information can be obtained by calculating cross-correlations between the points. Since some deep learning approaches achieved results by concatenating different numerical sequences such as different sensor signals and treating them as inputs, we can concatenate the original sensor signal with the cross-correlations. This defines the relative information, and treat it as an input. However it is not natural to treat them in the same way by simply concatenating them because the sensor signal represents a physical value and the cross-correlation represents their relationship. Hence, the essential meaning is different. We deal with these different values simultaneously by using a graph structure that represents each point and their relationships. Then, the obtained graph data is fed to a graph neural network for feature extraction. This enables the system to learn by focusing on the relative relationship of each coordinate of the data point.

The main reason for using a graph structure is to give data additional information. The time-frequency representation simply converts the original signals to other forms. Inspired by the success of CNN in computer vision, Wang & Oates (2015); Zhu et al. (2019) proposed encoding time series as different types of images using methods other than time-frequency analysis and inputting them into CNN. Umeda (2017) proposed a method of converting the original signals to high dimensional data cloud. While they are all categorized as information conversions, conversions meaning that they do not add any other information, we add relationship information as edges to the original signals, thus the graph hold richer information than the original signal and its conversions.

A key feature of the method is phase shift invariance. The application of our current research is for industrial machines with rotating shafts. Virtually every mechanized transportation vehicle, power generation, industrial machine, and robotic system contains rotating shafts. The shafts provide an opportunity to collect periodic signals. In practice, most measuring instruments such as sensors, processors and loggers along with their data acquisition systems show time delay respectively in the availability of the data. There is a limitation to correct the delays by hardware design or implementation. Hence, phase shifts may occur in the obtained periodic signals. However, these phase shifted signals are essentially the same. We identify them using a shift invariance method.

Our proposed method performs a cyclic permutation to a graph neural network. This method assures that the results account for phase shift of the periodic measurements. It is not necessary to consider the vertex order in the graph originally, but it is necessary to give the order for computability. Here, we identify the graphs whose vertex orders are different due to phase shifts. The conventional graph neural networks regard them different. Therefore, we propose a method that intentionally focuses on shift invariance by acting a cyclic permutation to a graph neural network. The use of this method in Section 3 shows that it offers predictions with sufficient accuracies for idealized data and the experimental data obtained from a test setup Gest et al. (2019).

## 2 MACHINE LEARNING METHODOLOGY

In this section, we define our machine learning method. First we introduce necessary terminology to review the graph theory. Then we describe the method of constructing a graph structure for each data point, and introduce the learning model corresponding to the graph data. Finally we extend the learning model to periodic data.

### 2.1 NOTATION AND TERMINOLOGY OF GRAPHS

The notation and terminology of graphs is as follows. Let $G$ be a graph. We denote the vertex set and the edge set of $G$ by the symbol $V(G)$ and $E(G)$. A *simple* graph is a graph containing no graph loops or multiple edges. A *complete graph* is a graph in which each pair of graph vertices is connected by an edge. An *ordered graph* is a graph with a total order over its vertices. If a graph

$G$ is ordered with $|V(G)| = N$, then we regard $V(G)$ as an ordered set $\{v_i\}_{1 \leq i \leq N}$. The *adjacency matrix* $A(G)$ of a simple ordered graph $G$ is a matrix with rows and columns labeled by graph vertices, with a 1 or 0 in position $(v_i, v_j)$ according to whether the vertices $v_i$ and $v_j$ are connected by an edge or not, respectively. For a graph $G$, a *vertex labeling* is a function from $V(G)$ to a set of labels. A graph with such a function defined is called a *vertex-labeled graph*. For a vertex-labeled graph $G$, we denote the vertex labeling by the symbol $\mathcal{L}(G)$. Unless otherwise stated we assume that graphs are simple ordered graphs labeled by real numbers.

## 2.2 Construction of Graph Structure

Now we construct a graph structure on each data point. Let $x$ be a data point, namely, $x$ be an $N$-dimensional real vector $(x_1, x_2, \ldots, x_N)$ for some integer $N$. For simplicity of notation, we write $(x_i)_{1 \leq i \leq N}$ instated of $(x_1, x_2, \ldots, x_N)$. Fix integers $ws$ and $ss$ with $1 \leq ws, ss \leq N$, we call these a *window size* and a *slide size*. Let $\lfloor \cdot \rfloor$ be the floor function, namely $\lfloor a \rfloor := \max\{n \in \mathbb{Z} \mid n \leq a\}$. Then we obtain the following $N'$-length sequence of $ws$-dimensional real vectors $\{v_i\}$, where $N' = \lfloor (N - ws)/ss \rfloor + 1$.

$$v_i = (x_{(i-1)*ss+1}, x_{(i-1)*ss+2}, \ldots, x_{(i-1)*ss+ws}) \text{ with } 1 \leq i \leq N'$$

We fix a real number $\varepsilon$ and a distance function $d$ on the $ws$-dimensional real vector space, such as the Euclidean distance or the correlation distance. Then we can define a graph $G$ as follows:

- $V(G) = \{v_i\}_{1 \leq i \leq N'}$,
- $E(G) = \{(v_i, v_j) \mid v_i, v_j \in V(G) \text{ with } d(v_i, v_j) < \varepsilon\}$,
- $\mathcal{L}(G) = \text{proj} : v_i \mapsto x_{(i-1)*ss+ws}$.

If the slide size and the window size are small enough, then maximum, minimum and mean work the same as a projection for the labeling function. However, in our case, vertices should be at least one period sub-waves because we construct edges by their similarity. Then max and min become constant values for every vertex. Also, periodically the window size becomes equal to a constant multiple of the period because our experimental data consists of many different frequencies. Then the mean becomes the same value for every vertex. Therefore, we use the projection as a vertex labeling in this paper.

## 2.3 Learning Model for Graph Data

Here we define our machine learning model suitable for graph data set. Our model draws inspiration from recent work on a graph neural network (Kipf & Welling, 2017). However, they consider the problem of classifying vertices of a graph by sharing filter parameters for each vertex and treat all vertices equal. On the other hand, since we consider graph-wise feature extraction and the vertices of our graph have time information as index, we think that it is not suitable to treat the vertices in the same way. Therefore, we avoid weight sharing by using the Hadamard product as shown below.

Set an integer $N$. Let $\mathcal{G}_N$ be a set of graphs with $N$-vertices. Set an integer $M$, which is a *number of hidden layers*. We take a finite sequence of $N$-by-$N$ matrices $\{W_m\}_{0 \leq m \leq M}$, called trainable *weights*, and a finite sequence of $N$-dimensional real vectors $\{b_m\}_{0 \leq m \leq M}$, called trainable *bias*. Then we define a function $\phi$ from $\mathcal{G}_N$ to $\mathbb{R}^N$ as follows:

$$\phi(G) = L_{M+1},$$
where, $L_0$ is the image of the labeling function $\mathcal{L}(G)(V(G))$,

$$L_{m+1} = \tau \left( L_m \cdot \left( \tilde{A}(G) \circ W_m \right) \right) + b_m.$$

Here, $\tilde{A}(G) = A(G) + I_N$ with the rank $N$ identity matrix $I_N$ and $\tau$ is an activation function, such as the ReLU, and $\cdot$ means the matrix product and $\circ$ means the Hadamard product. Graph neural networks usually use the Laplacian as $\tilde{A}(G)$, but our experimental results were almost the same, so we used $\tilde{A}(G) = A(G) + I_N$ instead. This turned out to be simpler than the Laplacian.

This function can be regarded as a natural extension of a multi-layer perceptron (MLP) obtained by admitting a graph structure on each data point. In fact, our function on the complete graph $G$, namely all the elements of $\tilde{A}(G)$ are 1, is equal to a MLP. Hence we call this function a *graph dynamic multi-layer perceptron* (GDMLP).

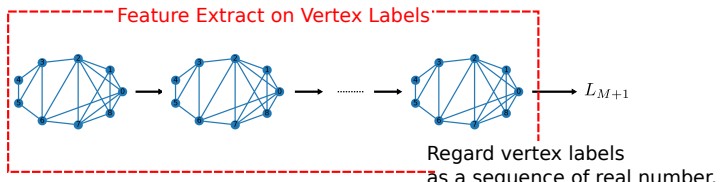

**Figure 1:** Graph dynamic multilayer perceptron (GDMLP).

## 2.4 EXTENSION OF LEARNING MODEL TO PERIODIC DATA

In this section, we extend the above learning model to another model which is suitable for periodic data. We consider its application to data obtained from industrial machineries containing rotating shafts. Accordingly each data point $x = (x_i)_{1 \leq i \leq N}$ obtained from them is *periodic* defined as follows. Let $T$ be the sampling interval, that is time between which data is recorded. Then $x = (x_i)_{1 \leq i \leq N}$ is *periodic* if there exists a *period* $T_x = n_x T$ with a positive integer $n_x$ such that $x_i = x_j$ if $i \equiv j \mod n_x$. On the other hand, it is difficult to perfectly control the collection timing of these data points because there is a limitation to correct the delays by hardware design or implementation. To account for this the data includes a *phase shift* $x' = (x'_i)_{1 \leq i \leq N}$ of $x$, namely, for periodic data points $x$ and $x'$ with a period $T_x = T_{x'} = nT$, there is a *delay* $T_{(x,x')} = n_{(x,x')}T$ with an integer $n_{(x,x')}$ such that $x_i = x'_{j - n_{(x,x')}}$ if $i \equiv j - n_{(x,x')} \mod n$. A period and a delay are multiples of the sampling interval $T$. This is a practical limitation and not a theoretical limitation. In general, sensor measurements are discrete in time. The sampling interval $T$ is the minimum interval when the sensor actually measures. Although we use this notation to clarify this practical limitation, sampling interval $T$ is not a limitation in the theoretical claim (see the appendix for details).

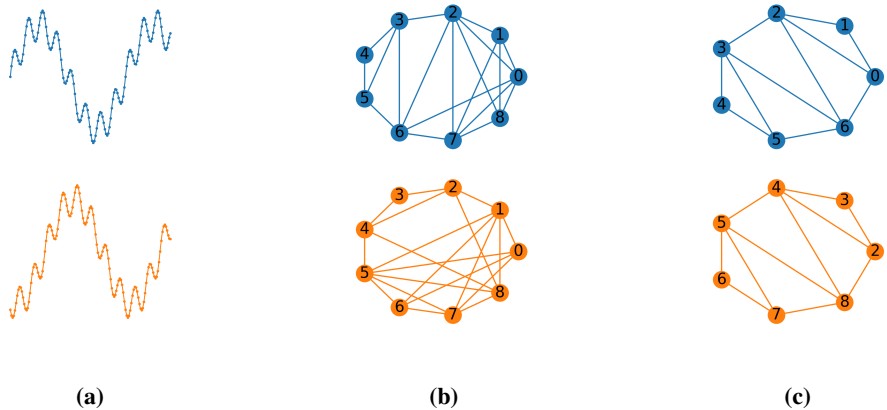

| (a) | (b) | (c) |

**Figure 2:** (a): Two waves with a phase shift. The above wave $x$ represented by $x_t = \sin \frac{2\pi t}{100} + 0.3 \sin \frac{16\pi t}{100}$ with $0 \leq t \leq 140$ and the below $x'$ is phase shifted wave of $x$ with 30 delay. (b): The graphs obtained from the left waves by the operation in Section 2.2 with $ws = 20$, $ss = 15$, euclidean distance as $d$ and $\varepsilon = 4.5$. (c): The subgraphs of the left graphs whose are equivalence up to cyclic order of vertices.

Although phase shift data points are different as vectors, the graphs obtained from them by the operation in Section 2.2 have "large" equivalent induced subgraphs up to cyclic order of vertices. To state more precisely we make the following definition; pick a $N$-vertices graph $G$. Let $\{v_i\}_{1 \leq i \leq N}$ be a vertex set $V(G)$. Let $\sigma$ be a cyclic permutation on $V(G)$ such that $\sigma(v_i) = v_{i+1}$. Then the *cyclic permutated graph* $\sigma(G)$ of $G$ is defined as follows:

- $V(\sigma(G)) = \{\sigma(v_i)\}_{1 \leq i \leq N}$,
- $E(\sigma(G)) = \{(\sigma(v_i), \sigma(v_j)) \mid (v_i, v_j) \in E(G)\}$,
- $\mathcal{L}(\sigma(G)) : \sigma(v_i) \mapsto \mathcal{L}(G)(v_i)$.

Let $x$ and $x'$ be phase shift data points. They are different as vectors (see Figure 2a). Hence the graphs $G_x, G_{x'}$ obtained from them by the operation in Section 2.2 are not equivalent as ordered graphs (see Figure 2b). However, most of them produce identical vertices to each other up to cyclic order (see Figure 2c). In fact we can prove the following claim.

**Claim .** Let $x$ and $x'$ be phase shifted data points with period $T_x = T_{x'} = nT$, and delay $T_{(x,x')} = n_{(x,x')}T$. Assume that $x$ and $x'$ is more than 3 periods, namely, $|x| = |x'| \geq kn$ for an integer $k \geq 3$. Then there exists a window size $ws$ and a slide size $ss$ satisfying the following condition. For graphs $G_x$ and $G_{x'}$ which obtained from $x$ and $x'$ by the operation in Section 2.2, there exists an integer $K_{(x,x')}$ such that both of $G_x$ and $\sigma^{K_{(x,x')}}(G_{x'})$ have a induced ordered subgraph $S_{(x,x')}$ satisfying $|V(S_{(x,x')})| \geq |V(G_x)|k/(k+1) = |V(G_{x'})|k/(k+1)$.

We provide a proof of the claim in the appendix. By the claim, it is expected that the GDMLP outputs of $G_x$ and $\sigma^{K_{(x,x')}}(G_{x'})$ will be approximately the same.

For the above reason, we improve a GDMLP with a cyclic order as shown in Figure 3. We fix a GDMLP model $\phi$ on $\mathcal{G}_N$. Then we define a function $\Phi_\phi$ from $\mathcal{G}_N$ to $\mathbb{R}^N$ as follows:

$$\Phi_\phi(G) = \rho\left(\phi(G), \phi(\sigma(G))), \phi(\sigma^2(G)), \ldots, \phi(\sigma^{N-1}(G))\right), \tag{1}$$

where, $\rho$ is a pooling function, such as the average-pooling. We call this function a *cyclic graph dynamic multilayer perceptron* (CGDMLP).

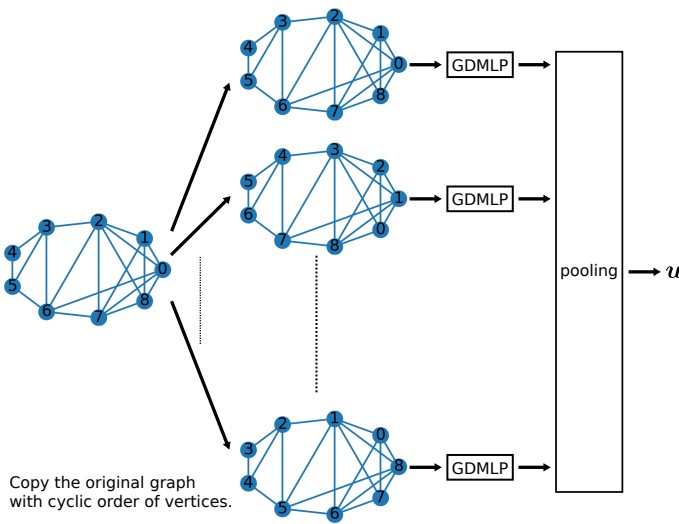

**Figure 3:** Cyclic graph dynamic multilayer perceptron (CGDMLP).

## 2.5 Learning Architecture

Since a GDMLP is a feature extraction of the graph vertex labels, it is necessary to compose a function with a GDMLP according to the final processing to be performed, such as a classification or a regression. Our main purpose is to extract similar features from phase shifted signals. In our experiments, we confirm the performance of the proposed feature extraction by checking the difference in accuracy with others. And this is independent of the classifier used. Therefore we compose a MLP with a GDMLP, which is a simple method. The MLP input layer consists of the same number of perceptrons as the dimension of the GDMLP output. Its output layer and the hidden layer is optimized according to the sophistication of the problem, such as the number of classifications desired. Similarly, in the case of a CGDMLP, we compose tailored a MLP according to the output desired (see Figure 4). For simplicity of notation, we use the same letter GDMLP and CGDMLP for the learning architecture whose feature extraction part are the feature extraction GDMLP and CGDMLP respectively.

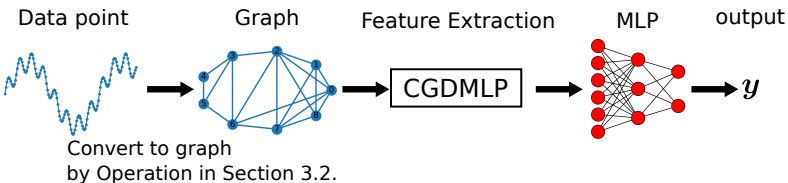

**Figure 4:** Learning Architecture.

# 3 EXPERIMENTS

We now give an example where we compare our method for analyzing periodic data.

## 3.1 IDEAL DATA

First, we apply our method to idealized data that abstracts the problem we are considering, such as a crack in a shaft, sensor malfunction, or external force that would impede mechanical rotation. The first of these would typically cause periodic noise, and the last two non-periodic noise. Reports of noises and their recurrence from a sensor, then, can be used to diagnose each of those and other issues. Our proposed method, which adds a graph structure to each data point to give a relative information, should effectively make such a diagnosis. Also, because our proposed CGDMLP is with a focus on cyclic order invariance, it is expected that phase shift signals can be identified because they produce approximately the same outputs in either unshifted or differently shifted signals.

Based on the above, we compare the classification result of each method for noisy sine waves $(x_t)_{1 \le t \le T}$ with some integer $T$ defined as follows:

$$x_t = \sin(2\pi f(t/T - t_0)) + \epsilon + \delta_f,$$

where $f$ is the frequency, $t_0$ is a phase shift, $\epsilon$ is a random noise from a uniform distribution over the half open interval $[-0.05, 0.05)$ and $\delta_f$ is periodic or non-periodic noise (see Figure 5).

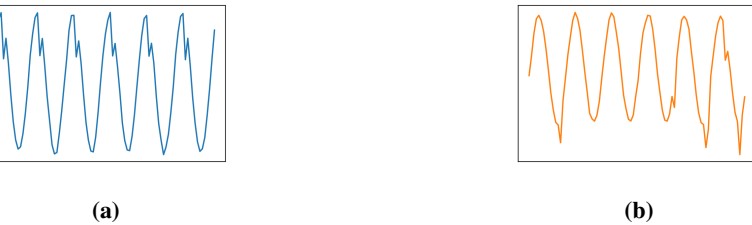

|     |     |
|:---:|:---:|
| (a) | (b) |

**Figure 5:** (a): Periodic noise, (b): Non-periodic noise.

Set the frequency $f$ to three types, 3, 6 and 9. We then execute $(x_t)_{1 \le t \le T}$ over the range of $\epsilon$ six times, once per each combination of frequency and periodic vs. non-periodic noise. Verification is performed with and without phase shift $t_0$. The validation set size for each result is fixed at 100. We train with several training set sizes, as shown in Table 1.

The architecture used for comparison is the architecture GDMLP and CGDMLP defined in Section 2.5. In addition, we use the architecture MLP and CMLP which are fully-connected networks obtained from GDMLP and CGDMLP by replacing all the elements of $\tilde{A}(G)$ with 1 respectively. We set the learning rate based on the LR range test (see Section 3.3 in Smith (2017)).

The results are summarized in Table 1. Our main purpose is to extract similar features from phase shifted signals. We focused on the difference in accuracy with other methods. In the middle and the bottom of Table 1 including phase shifted signals, when the training set size per class is 10, the difference in accuracy is higher than 30 %. It indicates that the proposed method can be performed with a small experimental dataset with phase shifts. On the other hand, the bottom third of Table 1 shows that the training data does not include phase shifted signals, however the proposed method performs acceptable with the validation set which includes unknown delays in signals. This could be considered as generalization performance when applied to phase shifted signals.

**Table 1:** Validation set accuracy for simple periodic data.

| Training Set Size Per Class | Training Phase Shift: None, Validation Phase Shift: None | | | |
|---|---|---|---|---|
| | MLP | CMLP | GDMLP | CGDMLP |
| 10 | 51.28 | 74.32 | 77.80 | 80.70 |
| 50 | 55.92 | 79.17 | 83.57 | 91.48 |
| 100 | 57.78 | 84.97 | 86.62 | 92.73 |

| Training Set Size Per Class | Training Phase Shift: Exist, Validation Phase Shift: Exist | | | |
|---|---|---|---|---|
| | MLP | CMLP | GDMLP | CGDMLP |
| 10 | 48.47 | 47.37 | 50.22 | 81.33 |
| 50 | 54.16 | 88.00 | 68.39 | 92.00 |
| 100 | 58.89 | 84.89 | 70.50 | 94.72 |

| Training Set Size Per Class | Training Phase Shift: None, Validation Phase Shift: Exist | | | |
|---|---|---|---|---|
| | MLP | CMLP | GDMLP | CGDMLP |
| 10 | 27.06 | 52.83 | 32.89 | 89.89 |
| 50 | 26.11 | 86.39 | 33.89 | 91.00 |
| 100 | 26.33 | 88.28 | 33.89 | 94.39 |

## 3.2 VALIDATION WITH REAL DATA

### 3.2.1 TEST SETUP

In order to validate our model on a real-world dataset, we use data obtained from the test setup Gest et al. (2019) as shown in Figure 6. The shaft is attached at both ends to brushed DC motors by compliant couplers. One motor, the driving motor, is connected to a power supply and electronic speed control and is controlled by a computer. The controls permit electrical current to the motor at any of five discrete voltage levels. The second motor, the damping motor, is attached to a resistor array to create a variable rotary damper. In the resistor array, relays are used connect and bypass individual resisters. The possible combinations allow for sixteen discrete levels of resistance. Five different weights of different masses and sizes are attached to the shaft to simulate the shaft bending, a possible real-world anomaly. Tests are also run and data collected with no weights attached. The sensors mounted on the shaft collect triaxial acceleration and strain, and audio data is collected with an external microphone. We collect a total of 400 cases of data from the above discrete variables. Each case consists of 50 data points. Each data point consists of 1.5 seconds signals. This time width is set to collect at least three rotations at the minimum rotation speed of 120 rpm. Due to the computational cost, each data point is downsampled to consist of 150 points.

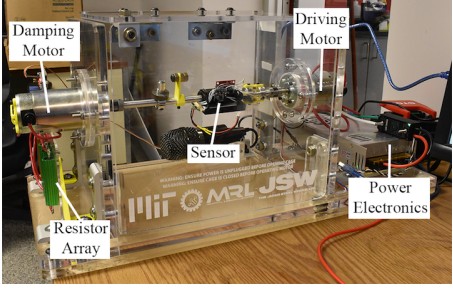

**Figure 6:** Experimental test setup.

### 3.2.2 CONSTRUCT MODELS FOR THE TEST SETUP DATA

To confirm the performance of the proposed feature extraction, we construct classification models to estimate the shaft bending. We use the measurements of tangential acceleration, radial acceleration and strain, as input by permutation importance (see Altmann et al. (2010), Breiman (2001)).

The basic architecture is the one defined in Section 2.5, but this time there are three types of sensor signals used for input, so each feature is extracted in parallel and the outputs concatenation is input to the multi-layer perceptron, which is a classifier (see Figure 7).

As in Section 3.1, for comparison we use GDMLP and CGDMLP, and MLP and CMLP which are fully-connected networks obtained from GDML and CGDMLP respectively.

In the preprocessing part each of the above three sensors data are converted to the graph data by the operation introduced in Section 2.2 with the following parameters. The window size $ws$ is set to $50$, which is set to include at least one period at minimum rotation speed of 120 rpm. The slide size $ss$ is set to 3, which is set to be smaller than one period at the maximum rotation speed of 1200 rpm. We set the distance function $d$ the correlation distance and $\varepsilon = 0.3$, namely, a pair of vertices are connected if they have a strong positive linear relationship.

We set the learning rates in this instance based on the LR range test as in Section 3.1 and we use a grid search optimization method for other parameters.

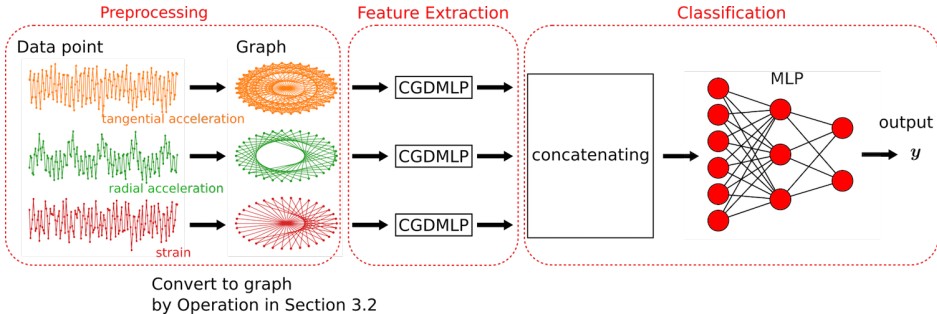

**Figure 7:** Learning Model for the Test Setup Data.

### 3.2.3 RESULTS

Table 2 shows the resulting accuracy of each method. There are only 60 samples in each case of the experimental data. For this limited data, our proposed method achieves sufficiently higher accuracy than either of the based methods MLP and CMLP. Of the proposed methods CGDMLP in particular achieves the highest accuracy. We suggest that our methods would be superior even with limited data since relative order information on the time axis was assigned here. We further suggest that CDGMLP, especially, would be superior even when evaluating data sets that include phase shift because our proposed CGDMLP is with a focus on cyclic order invariance.

**Table 2:** Result of each classification method.

| Model | Accuracy (%) |
|-------|--------------|
| MLP | 73.48 |
| CMLP | 71.62 |
| GDMLP | 80.96 |
| CGDMLP | 87.73 |

## 4 CONCLUSION

In this paper, we proposed a machine learning method for analyzing periodic data by admitting a graph structure to each data point and constructing a machine learning model according to the characters of the graph structure and the original data. Another point of importance is that adding a certain structure to data is shown to be very effective for feature extraction. The paper demonstrates experimentally the effectiveness of adding a graph structure to the data.

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

## A    APPENDIX

We give a proof of the claim in Section 2.4. First we recall the claim.

**Claim .** Let $x$ and $x'$ be phase shifted data points with period $T_x = T_{x'} = nT$, and delay $T_{(x,x')} = n_{(x,x')}T$. Assume that $x$ and $x'$ is more than 3 periods, namely, $|x| = |x'| \geq kn$ for an integer $k \geq 3$. Then there exists a window size $ws$ and a slide size $ss$ satisfying the following condition. For graphs $G_x$ and $G_{x'}$ which obtained from $x$ and $x'$ by the operation in Section 2.2, there exists an integer $K_{(x,x')}$ such that both of $G_x$ and $\sigma^{K_{(x,x')}}(G_{x'})$ have a induced ordered subgraph $S_{(x,x')}$ satisfying $|V(S_{(x,x')})| \geq |V(G_x)|k/(k+1) = |V(G_{x'})|k/(k+1)$.

*Proof.* If $n = 1$, it implies that $x$ and $x'$ are constant sequences. Thus we assume $n \geq 2$.

Let $n'_{(x,x')}$ be the remainder when $n_{(x,x')}$ is divided by $n$. By the definition of periodic and delay, we have $x_0 = x'_{n'_{(x,x')}}$ as follows:

$$
\begin{aligned}
x_0 &= x'_{n_{(x,x')}} && \text{(By the definition of delay.)} \\
&= x'_{qn + n'_{(x,x')}} && \text{(Here, q is the quotient when } n_{(x,x')} \text{ is divided by } n.) \\
&= x'_{n'_{(x,x')}} && \text{(By the definition of periodic.)}
\end{aligned}
$$

Similarly, we have $x'_0 = x_{n - n'_{(x,x')}}$. Note that $n'_{(x,x')} < n$ by the definition of the remainder. If $n'_{(x,x')} \geq n/2$, then $n - n'_{(x,x')} < n/2$. Thus, we can assume that there is an integer $m_{(x,x')} < n/2$ such that $x_{m_{(x,x')}} = x'_0$ by replacing $x$ and $x'$ if needed. Then $x$ and $x'$ have a continuous subsequence $s$ such that $|s| > (k-1)n$.

Set $ws < |s| - km_{(x,x')}$ and let $ss$ be a divisor of $m_{(x,x')}$. Note that we have $|s| - km_{(x,x')} > 1$ as follows:

$$
\begin{aligned}
|s| - km_{(x,x')} &> (k-1)n - k\frac{n}{2} && \text{By } |s| > (k-1)n \text{ and } m_{(x,x')} < n/2. \\
&= \left(\frac{k}{2} - 1\right) n \\
&\geq \left(\frac{3}{2} - 1\right) 2 && \text{By } k \geq 3 \text{ and } n \geq 2. \\
&= 1.
\end{aligned}
$$

Hence we can define $ws$ satisfying $ws < |s| - km_{(x,x')}$.

Since $ss$ is a divisor of $m_{(x,x')}$, for each $i > m_{(x,x')}/ss$, the vertex $v_i$ of $G_x$ is equivalent to some vertex $v'_j$ of $G_{x'}$. Let $S_{(x,x')}$ be the equivalent induced subgraphs consists of the above vertices. Since the graph $G_x$ is obtained by the operation in Section 2.2 and $ss$ is a divisor of $m_{(x,x')}$, we have $|V(G)| = |S_{(x,x')}| + m_{(x,x')}/ss$ and $|S_{(x,x')}| = \lfloor (|s| - ws)/ss \rfloor$. Since we set $ws < |s| - km_{(x,x')}$ and $ss$ is a divisor of $m_{(x,x')}$, we have $|S_{(x,x')}| \geq \lfloor km_{(x,x')}/ss \rfloor = km_{(x,x')}/ss$. Then we have $|S_{(x,x')}|/|V(G)| \geq k/(k+1)$ as follows:

$$
\begin{aligned}
|S_{(x,x')}|/|V(G)| &= \frac{|S_{(x,x')}|}{|S_{(x,x')}| + m_{(x,x')}/ss} && \text{(By } |V(G)| = |S_{(x,x')}| + m_{(x,x')}/ss.) \\
&= 1 - \frac{m_{(x,x')}/ss}{|S_{(x,x')}| + m_{(x,x')}/ss} \\
&\geq 1 - \frac{m_{(x,x')}/ss}{km_{(x,x')}/ss + m_{(x,x')}/ss} && \text{(By } |S_{(x,x')}| \geq km_{(x,x')}/ss.) \\
&= \frac{k}{k+1}.
\end{aligned}
$$

Then $m_{(x,x')}/ss$ is the desired integer $K_{(x,x')}$.

$\square$

