# OpenReview forum: "Cyclic Graph Dynamic Multilayer Perceptron for Periodic Signals"
_ICLR.cc/2020/Conference — Reject_

### Official Review · AnonReviewer4 · 2019-10-30
**Official Blind Review #4**

**Rating:** 3

**Review:**

This paper presents a novel architecture for extracting features for periodic signals that is sample efficient and has superior performance than previous approaches. The proposed method is based on a graph architecture that takes into account the ordering of the vertices, contrary to standard GNNs. In order to extract periodic signals, they prove that if two data points are phased shifted, then there exists a subgraph for each data point such that one is a cyclic permutation of the other. To that end, the authors present an architecture that is cyclic permutation invariant through a pooling operation on all the cycles. The proposed method is evaluated on simulated and real data.

The paper is well written and well motivated. The authors however ignore most of previous related work, there is a huge bulk of work on graph neural networks and on modelling time-series. It is important and interesting that the authors compare how the presented architecture differs from previous work.

Regarding the method, the entire approach is based on two assumptions: 1) the data points are phase shifted with a period that is a multiple of T, and 2) that you know that windows size and slide size such that makes one graph a cyclic permutation of the other. How often does (1) happen in practice? How sensitive it is to the failure of such assumptions?

The results section is the weakest part of this paper. The comparison between other approaches not presented by this method is essentially just the MLP, which is the most naive baseline. The author should compare to 1Dconvs, RNNs, MLPs with fourier features, and state-of-the-art approaches tackling time-series and/or periodic signals. As I mentioned previously, it would also be important to analyze the sensitivity of the method with respect to the assumptions build upon.

At this stage, I do not think that the paper is ready for acceptance.

**Experience Assessment:**

I have read many papers in this area.

**Review Assessment: Checking Correctness Of Derivations And Theory:**

I assessed the sensibility of the derivations and theory.

**Review Assessment: Checking Correctness Of Experiments:**

I assessed the sensibility of the experiments.

**Review Assessment: Thoroughness In Paper Reading:**

I read the paper at least twice and used my best judgement in assessing the paper.

---

### Official Review · AnonReviewer2 · 2019-10-31
**Official Blind Review #2**

**Rating:** 6

**Review:**

This paper proposes a machine learning framework for periodic data. The authors note that representing input data in vector form does not encode input coordinates relationship to one. Capturing this structure can be especially important for periodic signals. The authors address this by adding graph structure to each data point to encode relative structure about each coordinate. They then apply a graph neural network to the resulting structured data. They evaluate the method on an anomaly detection in a low data setting.

This paper would be greatly improved by an addition of a related work section. It is unclear where the novelty comes in precisely in this work because it is not very well situated within previous work on (i) anomaly detection w/ periodic signals, (ii) temporal and periodic signal processing and (iii) graph neural network approaches. Contextualized this work within these related areas would improve the clarity and readability of the work and also help frame the results.

The method is evaluated on synthetic and real datasets, comparing a couple variants of the model (one that can deal with phase shifts). Results support the main claims of the paper.

It is hard for me to assess the significance of this work since this is a very specific application of known techniques. I think the application is an important one, and also one that requires some domain knowledge, so there does appear to be a useful contribution in terms of adapting graph-based methodologies here. However, the methodology and application is outside my area of expertise.

Detailed suggestions / questions / comments:
- the ws-dimensional  vector v_i is defined as v_i = (x_{(i−1)∗ss+1}, x_{i+1}, . . . , x_{(i−1)∗ss+ws}). Is there a typo here? It's not obvious to me how the second index relates to the sequence or how this sequence is specified? Perhaps it should say  v_i = (x_{(i−1)∗ss}, x_{(i-1)*ss+1}, . . . , x_{(i−1)∗ss+ws})?
 - The authors mention alternative methods of capturing structured temporal information in the input features. For example, they suggest concatenating the original signal with the cross correlated signal. They also suggest time-frequency analysis methods (such as a Fourier transform and the wavelet transform) and applying a CNN to the time-frequency signal. They authors mention that these methods would require much more data than their graph convolution approach. I agree this is probably the case, but this would still be a useful empirical result to show the degree of data required for these alternative.
- What are previous approaches to detecting the properties explored in this work? In addition to discussing previous approaches in a related work section, some empirical analysis comparison would help contextualize this work as well.

Overall, I think this paper is a useful application of graph-based methods. The claims are verified empirically on real and synthetic data. I think it could be significantly improved with a discussion of related work and better situating of the methods / more comparisons in the results. As a result of these significant weaknesses I'm really on the fence with my recommendation -- the work is sensible but the paper has a lot of room for improvement and I'm not quite sure its ready for publication. However, it is possible underestimated the significance/impact of this work because I am not very familiar with the topic.

**Experience Assessment:**

I do not know much about this area.

**Review Assessment: Checking Correctness Of Derivations And Theory:**

I assessed the sensibility of the derivations and theory.

**Review Assessment: Checking Correctness Of Experiments:**

I assessed the sensibility of the experiments.

**Review Assessment: Thoroughness In Paper Reading:**

I read the paper at least twice and used my best judgement in assessing the paper.

---

### Official Review · AnonReviewer5 · 2019-10-31
**Official Blind Review #5**

**Rating:** 3

**Review:**

In the paper, the authors proposed a novel method adding graph architecture for collected data points to utilize not only features but also relative information. This helps reduce time and costs to collect a huge amount of data from industrial machines and improve accuracy. Although the paper idea is very interesting when presenting a new learning model for graph data, explanations and experiments are not convincing.

For explanations, the authors did not provide sufficient related works or references to prove that the problem the paper wants to solve is important. Also, for some approaches using deep learning mentioned there are no references.

For experiments, the results presented in Table 1 are good but there are no official baselines (e.g. from some prior works) to make the comparison more reliable.

Base on the arguments mentioned above, the paper is not convincing and reliable.

Small suggestion revision:
More analysis of prior works to show that the problem is important and need-to-solve
The introduction section should more references.
The experiments should be rigorous cause it lacks reliable baselines for comparison.


**Experience Assessment:**

I do not know much about this area.

**Review Assessment: Checking Correctness Of Derivations And Theory:**

I assessed the sensibility of the derivations and theory.

**Review Assessment: Checking Correctness Of Experiments:**

I assessed the sensibility of the experiments.

**Review Assessment: Thoroughness In Paper Reading:**

I read the paper at least twice and used my best judgement in assessing the paper.

---

### Official Review · AnonReviewer1 · 2019-11-03
**Official Blind Review #1**

**Rating:** 3

**Review:**

The goal of this work is to explore multi-sensor data modelling, with a focus on anomaly detection in machinery containing rotating shafts. Multi-sensor recordings from machinery can be phase-shifted, due to errors in the relative timing of sensors. The authors develop a method for modelling such phase-shift data. They augment phase-shift data with a graph structure which represents the relationship between sensors and they use a graph neural network with a cyclic permutation structure to enforce phase-shift invariance. Model performance is evaluated with real-world data from machinery containing a rotating shaft. Their model may be useful for other domains with phase-shifted data, such as multi-sensor medical data.

Although this is an interesting application domain and model, I have selected weak reject.

The primary reason for this decision is that the authors do not provide sufficient comparisons to related work and models, either in the form of a literature review, or in the form of model benchmarking. This is especially problematic for a domain which will be unfamiliar to much of the machine learning community.

None of the six references in the paper address anomaly detection for temporal data (e.g. Ahrens et al. “A machine-learning phase classification scheme for anomaly detection in signals with periodic characteristics” 2019) or the extensive related literature of time series models (e.g Pope et al. Learning phase-invariant dictionaries 2013, Edwards and Lee, Using Convolutional Neural Networks to Extract Shift-Invariant Features from Unlabeled Data”, 2019), or more closely related work on shift invariant graph neural networks (e.g. Gama et al, Convolutional neural network architectures for signals supported on graphs, 2018).

To address this, I feel that the authors need to provide a related work discussion. I would also like to see some experiments comparing their model to benchmark models that have a greater chance of being competitive such as a variant of models introduced by Pope et al. 2013 or Edwards and Lee 2019 for example.

Minor note: there is a typo in the definition of v_i

Thank you for the submission.


**Experience Assessment:**

I do not know much about this area.

**Review Assessment: Checking Correctness Of Derivations And Theory:**

I carefully checked the derivations and theory.

**Review Assessment: Checking Correctness Of Experiments:**

I carefully checked the experiments.

**Review Assessment: Thoroughness In Paper Reading:**

I read the paper at least twice and used my best judgement in assessing the paper.

---

### Decision · Program_Chairs · 2019-12-19

**Decision:**

Reject

**Comment:**

The reviewers all appreciated the area explored by this work but there was a consensus that it lacked a thorough presentation of existing works, as well as relevant baselines.

I encourage the authors to better position their work with respect to the existing literature for what should be a stronger submission for a future conference.